# The Regulation of Ferroptosis by Noncoding RNAs

**DOI:** 10.3390/ijms241713336

**Published:** 2023-08-28

**Authors:** Xiangnan Zheng, Cen Zhang

**Affiliations:** College of Biological Science and Engineering, Fuzhou University, Fuzhou 350108, China; xiangnanzheng@fzu.edu.cn

**Keywords:** ferroptosis, ncRNAs, iron metabolism, lipid peroxidation, miRNAs, lncRNAs, circRNAs

## Abstract

As a novel form of regulated cell death, ferroptosis is characterized by intracellular iron and lipid peroxide accumulation, which is different from other regulated cell death forms morphologically, biochemically, and immunologically. Ferroptosis is regulated by iron metabolism, lipid metabolism, and antioxidant defense systems as well as various transcription factors and related signal pathways. Emerging evidence has highlighted that ferroptosis is associated with many physiological and pathological processes, including cancer, neurodegeneration diseases, cardiovascular diseases, and ischemia/reperfusion injury. Noncoding RNAs are a group of functional RNA molecules that are not translated into proteins, which can regulate gene expression in various manners. An increasing number of studies have shown that noncoding RNAs, especially miRNAs, lncRNAs, and circRNAs, can interfere with the progression of ferroptosis by modulating ferroptosis-related genes or proteins directly or indirectly. In this review, we summarize the basic mechanisms and regulations of ferroptosis and focus on the recent studies on the mechanism for different types of ncRNAs to regulate ferroptosis in different physiological and pathological conditions, which will deepen our understanding of ferroptosis regulation by noncoding RNAs and provide new insights into employing noncoding RNAs in ferroptosis-associated therapeutic strategies.

## 1. Introduction

Iron is an essential trace element for virtually all living organisms. It plays important roles in many physiological and cellular processes, including oxygen transport, energy production, and cellular proliferation [1,2]. Because of its importance, iron levels are finely tuned in living organisms, and iron overload can damage an organism through a variety of mechanisms, including the induction of a kind of regulated cell death, ferroptosis. The term “ferroptosis” was first introduced in 2012, which refers to a type of regulated cell death resulting from iron overload and lipid peroxidation [3,4,5]. Therefore, it cannot be blocked by the specific inhibitors for other regulated cell death types, such as apoptosis, necrosis, necroptosis, and autophagy, but can be blocked by iron chelators (e.g., deferoxamine mesylate) and lipid peroxidation inhibitors (e.g., ferrostatin-1 and liproxstatin-1) [3,6,7]. As a novel form of regulated cell death, ferroptosis is regulated by its unique metabolism and regulatory mechanism, leading to different morphological, biochemical, and immunological features from apoptosis, autophagy, necroptosis, and pyroptosis [4,8]. Normally, ferroptotic cells exhibit morphological changes mainly in their mitochondria, including obviously smaller mitochondria, rupture of the mitochondrial outer membrane, and reduced or absent mitochondrial crista [3,5,6]. They also show necrosis-like features, such as plasma membrane integrity loss, cytoplasmic and cytoplasmic organelles swelling, and moderate chromatin condensation. However, there are no apoptotic bodies or autophagosomes in ferroptotic cells, which are signature morphological features in apoptosis and autophagy, respectively. Iron accumulation and lipid peroxidation are the main biochemical features of ferroptosis, both of which are related to the mechanism of ferroptosis [4,5,6]. Additionally, ferroptosis is considered a form of inflammatory cell death immunologically, which is associated with the release of damage-associated molecular pattern (DAMP) or lipid oxidation products [4]. Since it was discovered, increasing evidence has shown that ferroptosis plays special roles in many physiological and pathological processes, such as cancer, neurodegeneration diseases, cardiovascular diseases, and ischemia/reperfusion injury (IRI) [9,10,11,12,13,14].

Noncoding RNAs (ncRNAs) refer to the RNA molecules transcribed from genomes that do not encode proteins. With the advent of powerful sequencing technologies and in silico tools, it was found that up to 90% of genes in eukaryotic genomes have the capacity to be transcribed into RNAs [15,16,17]. Only 1–2% of the genes encode proteins, whereas the majority of the genes are transcribed as noncoding RNAs (ncRNAs). According to their length and shape, ncRNAs are divided into various types, including transfer RNAs (tRNAs), ribosomal RNAs (rRNAs), microRNAs (miRNAs), small interfering RNAs (siRNAs), circular RNAs (circRNAs), PIWI-interacting RNAs (piRNAs), small nuclear RNAs (snRNAs), small nucleolar RNAs (snoRNAs), long noncoding RNAs (lncRNAs), etc. Different from the message RNAs (mRNAs) that encode proteins, ncRNAs play important roles in transcriptional and posttranscriptional levels as well as in the epigenetic regulation of gene expression [18,19]. With in-depth research, ncRNAs have been found to participate in multiple biological processes, including both physiological and pathological processes. In the past several years, more and more ncRNAs have been reported to be involved in ferroptosis, which adds a complex network for the regulation of ferroptosis in different physiological and pathological conditions. In this review, we summarize the recent advances in the mechanism of ferroptosis, especially the role of ncRNAs, including the function of different types of ncRNAs in ferroptosis and the mechanism for ncRNAs regulating ferroptosis.

## 2. The Core Mechanism and Regulators of Ferroptosis

Ferroptosis is an iron-dependent, oxidative-damage-related type of regulated cell death [4,5]. Although ferroptosis has been studied for more than ten years, our knowledge of its mechanism is still limited. An increased iron accumulation and lipid peroxidation can induce ferroptosis, whereas the antioxidant defense systems inhibit lipid peroxidation and ferroptosis [4,5,6]. Generally, it is universally accepted that ferroptosis is regulated by iron metabolism (leading to abnormal iron accumulation), lipid metabolism (leading to increased lipid peroxidation), and the antioxidant defense systems (leading to the dysregulation of antioxidant defense) (Figure 1). Additionally, many proteins, especially transcription factors, regulate ferroptosis by directly or indirectly modulating iron metabolism, lipid metabolism, and the antioxidant defense systems.

### 2.1. Iron Metabolism

Ferroptosis is a form of regulated cell death dependent on iron, which exchanges between two states, ferrous (Fe^2+^) and ferric (Fe^3+^), in cells. Indeed, only intracellular free Fe^2+^ (but not Fe^3+^) induces ferroptosis. Fe^2+^ induces lipid peroxidation and ferroptosis in two ways, by directly generating ROS by the Fenton reaction or by activating iron-containing enzymes that catalyze lipid peroxidation (e.g., lipoxygenase) [5,11]. The absorption, storage, efflux, and utilization of iron, which regulate intracellular free Fe^2+^ levels, may affect ferroptosis [5,11].

Dietary non-heme iron is absorbed by proximal small intestinal mucosal absorptive cells and is then transported to the blood, where it is oxidized to Fe^3+^ and binds to transferrin (TF) [20]. TF with Fe^3+^ is recognized by the transferrin receptor (TFRC) in the cell membrane, and the Fe^3+^-TF-TFRC complex comes into the cytoplasm through endocytosing [20,21]. In the endosome, Fe^3+^ is reduced to Fe^2+^ by the ferrireductase activity of six-transmembrane epithelial antigens of the prostate 3 (STEAP3) [22]. The reduced Fe^2+^ is further released into a free iron pool in the cytoplasm by divalent metal transporter 1 (DMT1, also known as solute carrier family 11 member 2, SLC11A2) [20,21]. Elevation of TF, TFRC, STEAP3, or DMT1 was found to promote ferroptosis by enhancing iron uptake [5,21]. It was also found that heat shock protein family B (small) member 1 (HSPB1), when phosphorylated at serine 15 by protein kinase C (PKC), inhibits the iron uptake process by regulating the cytoskeleton organization, resulting in the suppression of ferroptosis [23]. Heme iron is taken up as heme by receptor-mediated endocytosis or heme transporters [24]. Internalized heme is then degraded by heme oxygenase 1 (HO-1) or HO-2, releasing the Fe^2+^ to the cytoplasm. Both HO-1 and HO-2 were reported to be ferroptosis regulators [25,26]. Two mitochondrial iron importers, mitoferrin-1 (also known as solute carrier family 25 member 37, SLC25A37) and mitoferrin-2 (also known as solute carrier family 25 member 28, SLC25A28), were found to facilitate ferroptosis by increasing mitochondrial free iron accumulation [27,28].

To maintain the free iron hemostasis in cells, excess intracellular Fe^2+^ is stored to avoid iron overload. Ferritin is a cytosolic iron-storage protein complex comprising 24 subunits of two types, ferritin-light chains (FTL) and ferritin-heavy chains 1 (FTH1) [21]. Increasing the expression of *FTL* and *FTH1* will reduce intracellular free Fe^2+^ and inhibit ferroptosis [21]. When ferritin binds to nuclear receptor coactivator 4 (NCOA4), it is transported to autophagosome for lysosomal degradation (known as ferritinophagy), releasing ferritin-bound iron in order to increase intracellular free Fe^2+^ levels and result in ferroptosis [29]. DNA (cytosine-5)-methyltransferase 1 (DNMT-1) can enhance NCOA4 expression via DNA methylation in the NCOA4 promoter, leading to NCOA4-mediated ferritinophagy and ferroptosis [30]. Poly(RC)-binding proteins (PCBPs) act as iron chaperones to deliver Fe^2+^ to different proteins [21]. It was found that PCBP1 and PCBP2 deliver Fe^2+^ to ferritin, resulting in the suppression of ferroptosis [31,32]. PCBP2 was also reported to inhibit ferroptosis by transporting Fe^2+^ to ferroportin [33].

Iron-efflux protein ferroportin (FPN1, also known as solute carrier family 40 member 1, SLC40A1) is responsible for the exportation of intracellular free Fe^2+^ and the re-oxidation of Fe^2+^ to Fe^3+^ [34]. Hepcidin (encoded by the *Hamp1* gene) can induce the internalization and degradation of FPN1 [35]. A decrease in FPN1 or an increase in hepcidin levels can promote ferroptosis by increasing intracellular free Fe^2+^ levels [21]. Prominin 2 (PROM2), a pentaspanin protein involved in the organization of plasma membrane microdomains, was also found to export iron by promoting a form of ferritin-containing multivesicular bodies, resulting in the resistance of ferroptosis [36,37]. Iron is also used for iron-sulfur cluster biogenesis. The expression of some mitochondrial proteins with an iron-sulfur cluster, such as NFS1, CISD1, and CISD2, may reduce the available iron levels and thus inhibit ferroptosis [38,39,40].

Moreover, iron-regulatory proteins (IRPs), including IRP1 (also known as aconitase 1, ACO1) and IRP2 (also known as iron-responsive element binding protein 2, IREB2), register cytosolic iron concentrations and post-transcriptionally regulate the expression of iron metabolism genes to optimize cellular iron availability [41,42]. IRPs can bind to iron-responsive elements (IREs), the specific RNA stem–loop structures located in 5′- or 3′- untranslated regions (UTR) of mRNA, leading to the opposite effect: the inhibition of translation at 5′- UTR or the promotion of translation at 3′- UTR. IRPs can bind to the mRNA of *TFRC*, *DMT1*, *FTH1*, *FTL*, and *FPN*, affecting ferroptosis sensitivity [3,21,43].

### 2.2. Lipid Metabolism

Ferroptosis directly results from lipid peroxidation, which occurs in polyunsaturated fatty acids (PUFAs) through both non-enzymatically spontaneous autoxidation and enzyme-mediated processes [44,45]. Lipoxygenases (LOXs) are the major enzymes responsible for enzymatical lipid peroxidation by catalyzing the oxygenation of PUFAs to generate various hydroperoxy PUFA derivatives, including the initial lipid hydroperoxides and the subsequent reactive toxic aldehydes (e.g., malondialdehyde and 4-hydroxynonenal). The inhibition of LOX activities or reducing their levels inhibit ferroptosis [46]. The arachidonate lipoxygenase (ALOX) family, a class of non-heme-iron-containing LOXs, play a tissue- or cell-dependent role in mediating ferroptosis [44,45]. Additionally, cyclooxygenase-2 (COX-2, encoded by the PTGS2 gene) oxidizes lysophospholipids but not phospholipids; therefore, it was considered as a biomarker but not a driver for ferroptosis in early studies [5,47]. However, recent studies have found that COX-2 can mediate lipid peroxidation and induce ferroptosis in certain conditions [48,49]. Alternatively, non-enzymatic lipid peroxidation requires free radicals, including ROS. Some membrane electron transfer proteins, including nicotinamide adenine dinucleotide phosphate (NADPH) oxidases (NOXs) and cytochrome P450 oxidoreductases (PORs), contribute to ROS production and thus induce ferroptosis [50,51,52]. 

Lipid peroxidation occurs in PUFAs, which are supplied by the lipid synthesis and metabolism pathways. Ferroptosis results from the peroxidation of PUFA-phospholipids in the membrane [5,45]. Two PUFAs, arachidonic acid (AA) and adrenic acid (AdA), are the main substrates in PUFA-phospholipids during ferroptosis [44,45]. AdA/AA is ligated with CoA by Acyl-CoA synthetase long-chain family member 4 (ACSL4) to form CoA-AdA/AA, which then undergoes esterification with membrane phosphatidylethanolamine by lysophosphatidylcholine acyltransferase 3 (LPCAT3) to form PE-AdA/AA. A deficiency of either ACSL4 or LPCAT3 leads to the suppression of ferroptosis [53]. On the contrary, because of the lack of the bis-allylic positions readily available for peroxidation, monounsaturated fatty acids (MUFAs) in membrane phospholipids may inhibit ferroptosis by competing with PUFAs. Similarly, MUFAs are converted to MUFA-CoAs by Acyl-CoA synthetase long-chain family member 3 (ACSL3) and then form PE-MUFAs by membrane-bound O-acyltransferase domain containing 1/2 (MBOAT1/2). Therefore, promoting ACSL3 or MBOAT1/2 expression inhibits ferroptosis by elevating PE-MUFAs [54,55,56].

Long-chain PUFAs, including AA and AdA, are synthesized from dietary essential fatty acids (e.g., linoleic acid) by a series of enzymatic reactions in cells. These reactions are catalyzed by the elongation of very-long-chain fatty acid proteins (ELOVLs) and fatty acid desaturases (FADSs). It was reported that the inhibition or silencing of these enzymes, such as ELOVL5, FADS1, and FADS2, can repress ferroptosis [57,58]. Different from PUFAs, MUFAs come from saturated fatty acids (SFAs) produced by stearoyl-CoA desaturase 1 (SCD1), whereas SFAs can be de novo synthesized from acetyl-CoA by acetyl-CoA carboxylase (ACC) and fatty acid synthase (FASN). Increasing the expression of SCD1, FASN, and ACC protects cells from ferroptosis [55,59,60,61]. Alternatively, dietary fatty acids are another important source of PUFAs, MUFAs, and SFAs, especially for cells incapable of synthesizing fatty acids. Fatty acid translocase (FAT/CD36), fatty acid transport proteins (FATPs), and fatty-acid-binding proteins (FABPs) mediate the uptake of fatty acids [62]. It was reported that CD36 and FATP2, which mediate the absorption of AA and AdA, promote ferroptosis in certain cells [63,64]. FABP3, FABP4, and FABP7 were also reported to inhibit ferroptosis by regulating fatty acid uptake [55,65].

### 2.3. Antioxidant Defense Systems

Different from iron metabolism and lipid metabolism, the antioxidant defense system in cells suppresses lipid peroxidation, leading to the inhibition of ferroptosis. A selenocysteine-containing enzyme of the antioxidant defense system, glutathione peroxidase 4 (GPX4), plays a central role in ferroptosis inhibition. GPX4 is the only known enzyme that directly inhibits lipid peroxidation by using glutathione (GSH) to reduce toxic phospholipid hydroperoxides (PL-OOH) to non-toxic phospholipid alcohols (PL-OH) [66,67]. Therefore, its dysfunction always leads to the accumulation of lipid peroxides and ferroptosis. Many small-molecule ferroptosis inducers, such as RSL3, ML162, ML210, FIN56, and FINO2, induce ferroptosis by inhibiting GPX4 activity or promoting its degradation [66,67,68,69]. Interestingly, GPX4 can be regulated by lipid metabolism. GPX4 is a selenocysteine-containing protein, which needs selenocysteine tRNA for its synthesis. Selenocysteine tRNA comes from isopentenyl pyrophosphate (IPP), a product of the mevalonate pathway for lipid synthesis [70]. Thus, GPX4 levels are regulated by the mevalonate pathway via selenocysteine tRNA. Moreover, it was also reported that GPX4 is regulated by SCD1/FADS2, two important enzymes involved in lipid metabolism [71]. 

GPX4 uses GSH to reduce PL-OOH to PL-OH; therefore, the availability of GSH is also important for the inhibition of lipid peroxidation and ferroptosis by GPX4. GSH is synthesized from glutamate, cysteine, and glycine by glutamate cysteine ligase (GCL) and glutathione synthetase (GSS), whereas ChaC glutathione-specific γ-glutamyl-acyltransferase 1 (CHAC1) cleaves GSH into 5-oxo-L-proline and a Cys-Gly dipeptide. The inhibition of GCL or GSS, or activating CHAC1, may reduce GSH levels and induce ferroptosis [3,68,72,73,74]. Moreover, amino acid metabolism, which regulates the supply of glutamate, cysteine, or glycine, also plays crucial roles in ferroptosis. Among the metabolism of amino acids, the uptake of cystine, which can be further converted into cysteine by GSH and/or thioredoxin reductase 1 (TXNRD1), is considered to be a key mechanism for inducing ferroptosis [75]. Many small-molecule compounds, including erastin, sulfasalazine, and sorafenib, trigger ferroptosis by inhibiting system Xc^−^, a glutamic acid/cystine antiporter in the plasma membrane responsible for the uptake of cystine [68,75,76]. The suppression of the expression levels of its two subunits, solute carrier family 7 member 11 (SLC7A11) and solute carrier family 3 member 2 (SLC3A2), also exhausts GSH and induces ferroptosis [77,78]. Cysteine is also synthesized through the trans-sulfuration pathway by cysteinyl-tRNA synthetase (CARS) and cystathionine-β-synthase (CBS), leading to resistance to ferroptosis [79,80,81,82]. Additionally, glutaminolysis, the catabolism of glutamate and glutamine, is another target for the regulation of ferroptosis. Glutamine is absorbed mainly by glutamine transport, solute carrier family 1 member 5 (SLC1A5), and solute carrier family 38 member 1 (SLC38A1) [83,84]. It is converted into glutamate by glutaminases, and glutamate can be further converted into α-ketoglutarate by glutamic oxaloacetic transaminase (GOT). Glutamate can help the system-Xc^−^-mediated absorption of cystine and can supply the GSH synthesis directly. Some studies showed that SLC1A5 and SLC38A1 suppress ferroptosis [85,86,87], whereas GOT1 promotes ferroptosis through elevating GSH levels [88]. However, more studies found that SLC1A5, SLC38A1, and glutaminase 2 (GLS2) promote glutaminolysis and ferroptosis [83,89,90,91,92,93,94,95,96]. These were attributed to the enhancement of α-ketoglutarate, which promotes ferroptosis probably by increasing lipid synthesis, the local iron level, and mitochondrial ROS [83,97,98]. It seems that glutaminolysis has a context-dependent dual function in ferroptosis. 

In addition to the GPX4-GSH system, many other members of the antioxidant defense system were also found to regulate ferroptosis. Ferroptosis-suppressor protein 1 (FSP1, previously known as apoptosis-inducing factor mitochondria-associated 2, AIFM2) can trap lipid peroxides using NADPH to reduce non-mitochondrial coenzyme Q10 (CoQ), thereby repressing ferroptosis in a GPX4-independent manner [99,100]. Moreover, FSP1 was also reported to activate the ESCRT-III-dependent membrane repair system to inhibit ferroptosis, which is independent of its oxidoreductase function [101]. Tetrahydrobiopterin (BH_4_) is another radical-trapping antioxidant that protects lipid membranes from ferroptosis, which prevents two PUFA acyl tails from consuming phospholipids [102]. Both GTP cyclohydrolase-1 (GCH1) and dihydrofolate reductase (DHFR) can inhibit ferroptosis by producing BH_4_ [102,103]. In mitochondria, dihydroorotate dehydrogenase (DHODH) reduces CoQ to radical-trapping antioxidant ubiquinol (CoQH2), leading to the restoration of peroxide-damaged mitochondrial lipids and the inhibition of ferroptosis [104]. Additionally, several peroxiredoxins (PRDXs), members of a selenium-independent glutathione peroxidase family, were reported to inhibit ferroptosis [105,106,107]. It was also found recently that tryptophan can produce two radical-trapping antioxidants, serotonin (5-HT) and 3-hydroxyanthranilic acid (3-HAA), to eliminate lipid peroxidation, thereby inhibiting ferroptosis [108].

### 2.4. Transcription Factors Regulating Ferroptosis

Many transcription factors, as well as their related signaling pathways, are also involved in the regulation of ferroptosis by directly or indirectly affecting the above-mentioned mechanisms. Nuclear factor erythroid 2-related factor 2 (NRF2, also known as nuclear factor erythroid-derived 2-like 2, NFE2L2) is a master regulator of oxidative stress signaling and redox homeostasis [109,110]. NRF2 transcriptionally regulates a group of genes involved in antioxidant defense, such as *SLC7A11*, *TXNRD1, GSS, GCLC*, *GCLM*, *CHAC1*, *GPX4*, *FSP1*, *ARK1C1*, *ALDH1A1*, and *NQO1*, to inhibit ferroptosis. It was also reported that NRF2 can suppress ferroptosis by transcriptionally regulating the expression of *FPN1*, *HO-1*, *FTL*, *FTH1*, *ABCB6*, *FECH*, and *HRG1* (*SLC48A1*) involved in iron metabolism and *PPARG* and *NROB2* involved in lipid metabolism [109,110]. Sterol regulatory-element binding proteins (SREBPs) are transcription factors that regulate the expression of genes involved in lipid synthesis [111,112]. SREBP1 is the master regulator of lipogenesis through transcriptionally inducing the expression of *ACC*, *FASN*, and *SCD1*. The knockdown of *SREBP1* or *SCD1* will sensitize cancer cells to ferroptosis [55,61]. Signal transducer and activator of transcription 3 (STAT3), a transcription-factor-regulating gene associated with cell survival, the cell cycle, and immune reaction, was found to inhibit ferroptosis by increasing the expression of *SLC7A11* and *GPX4* for antioxidant defense, reducing *ACSL4* expression for lipid metabolism, and increasing hepcidin expression for iron metabolism [113,114,115,116].

Some transcription factors have a dual function in ferroptosis. Tumor suppressor p53 is a transcription factor that mediates a variety of anti-proliferative processes via transcriptionally regulating many stress-response genes [117]. Normally, the activation of p53 induces ferroptosis by transcriptionally regulating *SLC7A11*, *GLS2*, *CBS*, and *lncRNA-LINC00336*/miR-6852/CBS involved in the anti-oxidant defense, *ALOX12*, *SAT1*/ALOX15, *PTGS2*, and *iPLA2β* involved in lipid metabolism, and *FDXR*, *mitoferrin-2*, and *lncRNA-PVT1*/miR-214/TFRC involved in iron metabolism [117,118]. p53 was also reported to be a ferroptosis suppressor by enhancing GSH levels via its target, *p21*, inducing mitophagy via *Parkin* and inhibiting NOX-mediated lipid peroxidation via directly binding the dipeptidyl peptidase DPP4 [117,118]. Thus, p53 regulates ferroptosis in a bidirectional and context-dependent way. Hypoxia-inducible factors (HIFs) are transcription factors that respond to decreases in available oxygen (hypoxia). HIF-1 and HIF-2, two HIFs mediating most of the cellular response to hypoxia, transcriptionally regulate *TFRC*, *FTMT*, and *CA9* for iron metabolism, *FABP3*, *FABP7*, and *lncRNA-CBSLR*/CBS/ACSL4 for lipid metabolism, and *PDK1*, *BNIP3*, *METTL14*/SLC7A11, and *lncRNA-PMAN*/ELAVL1/SLC7A11 for antioxidant defense, eventually inhibiting ferroptosis in certain conditions, whereas under some different conditions, they can also transcriptionally regulate *TF, TFRC, DMT1, ZIP8*, and *ZIP14* for iron metabolism, *ACSL4, PTGS2*, and *HILPDA* for lipid metabolism, and *CHAC1* and *SOD* for antioxidant defense, leading to the promotion of ferroptosis [119]. Some activated transcription factors (ATFs), such as ATF3 and ATF4, which are activated in response to ER stress, also play dual roles in ferroptosis. Both ATF3 and ATF4 can transcriptionally induce *SLC7A11* to inhibit ferroptosis [120,121,122,123,124]. It was also reported that ATF3 represses brucine-induced glioma cell ferroptosis by upregulating *NOX4* and *SOD1* to reduce ROS [125] and inhibits IRI-induced cardiomyocyte ferroptosis by transcriptionally inducing *Fanconi anemia complementation group D2* (*FANCD2*) to affect the expression of *GPX4*, *SLC7A11*, *FTH1*, and *PTGS2* [126]. ATF4 was also found to transcriptionally induce *HSPA5*, which in turn binds GPX4 and protects against GPX4 protein degradation, resulting in the suppression of ferroptosis [127,128]. On the other hand, both ATF3 and ATF4 can transcriptionally induce *CHAC1* to reduce GSH and promote ferroptosis [72,129,130]. *ATF3* is transcriptionally regulated by ATF4 to upregulate *TFRC* expression to promote ferroptosis [131]. ATF3 was also found to promote IFN-γ-driven ferroptosis by increasing the transcription of an miRNA *miR-21-3p* to repress TXNRD1 [132], whereas ATF4 induces *B-cell translocation gene 1* (*BTG1*) to enhance the ferroptosis of hepatocytes [133].

Taken together, the mechanisms of ferroptosis are very complex. The core mechanisms of ferroptosis are iron accumulation, lipid peroxidation, and antioxidant defense, which are further regulated by many transcription factors and other proteins.

## 3. The Classification and Function of ncRNAs in Ferroptosis

ncRNAs can be divided into two subtypes, basic structural ncRNAs and regulatory ncRNAs, according to their function [134]. The former are also known as “housekeeping” ncRNAs, including tRNAs, rRNAs, snRNAs, and snoRNAs, which are constitutively expressed and function during the translation and splicing process. The regulatory ncRNAs contain miRNAs, siRNAs, circRNAs, piRNAs, and lncRNAs, which play important roles in the epigenetic regulation of gene expression. At present, the majority of ncRNAs involved in ferroptosis regulation are the regulatory ncRNAs, especially miRNAs, circRNAs, and lncRNAs.

### 3.1. MiRNAs in Ferroptosis

MiRNAs are small single-stranded ncRNAs of from 21 to 23 nucleotides derived from endogenous short-hairpin transcripts [135,136]. MiRNAs normally bind to complementary sequences within the 3′-UTRs of target mRNAs, leading to the cleavage, degradation, or translation inhibition of target mRNAs and eventually suppressing their expression [136,137]. 

In recent years, many studies have reported that miRNAs can target the ferroptosis-related genes to regulate ferroptosis. For instance, miRNA-214 (miR-214) targets and inhibits TFRC to reduce ferroptosis [138]; miR-424-5p and miR-4291 target and inhibit ACSL4 to suppress ferroptosis [139,140]; miR-30b-5p and miR-124 target and inhibit FPN1 to promote ferroptosis [141,142]; miR-541-3p and miR-324-3p target and inhibit GPX4 to enhance ferroptosis [143,144,145]; and miR-128-3p, miR-375, and miR-27a-3p target and inhibit SLC7A11 to induce ferroptosis [146,147,148]. 

In addition to mRNA, miRNA-binding sequences also exist in some other ncRNAs. These ncRNAs, including RNAs of pseudogenes, lncRNAs, and circRNAs, can compete with mRNA for the same miRNA pool, thereby regulating miRNA activity, which adds a level of regulation to the miRNA network [149,150]. 

### 3.2. LncRNAs in Ferroptosis

LncRNAs are a large group of RNAs, generally defined as transcripts of more than 200 nucleotides that are not translated into proteins [151,152]. LncRNAs regulate gene expression in a variety of ways at the epigenetic, including chromatin remodeling, transcriptional, translational, and post-translational regulations [153]. 

Some lncRNAs bind to miRNAs in a competitive manner as a miRNA sponge, leading to the inhibition of miRNAs. Therefore, some of these lncRNAs can regulate ferroptosis by inhibiting ferroptosis-related miRNAs. LncRNA PVT1 (lncPVT1) was found to inhibit miR-214 and enhance TFRC and p53 to induce ferroptosis in brain ischemia/reperfusion (I/R) [138]. LncOIP5-AS1 inhibits miR-128-3p to elevate SLC7A11 expression, eventually inhibiting ferroptosis in prostate cancer [146]. 

In addition, lncRNAs were also found to regulate ferroptosis by interacting with proteins to modulate mRNA stability and protein ubiquitination. For example, lncPMAN can bind to ELAVL1, leading to the stabilization of *SLC7A11* mRNA to inhibit ferroptosis [154], whereas lncHEPFAL can promote SLC7A11 protein ubiquitination and degradation to promote ferroptosis [155]. Moreover, lncP53RRA can interact with Ras GTPase-activating protein-binding protein 1 (G3BP1) to sequester p53 in the nucleus, thus promoting ferroptosis [156]. 

### 3.3. CircRNAs in Ferroptosis

CircRNAs are a class of single-stranded RNAs that form a closed continuous loop from the 3′ to 5′ end, which are highly expressed in the eukaryotic transcriptome [157,158,159]. Unlike linear RNAs, circRNAs have covalently closed circular structures without a 5′ cap structure and a 3′ polyA tail and are derived from exons via alternative mRNA splicing. CircRNAs are much more stable than linear RNAs [158,160,161,162], which have a high degree of stability and a potential effect on gene regulation [163,164,165]. 

CircRNAs usually also act as competitive RNA or RNA sponges to bind miRNAs, thus regulating the target proteins of those miRNAs. In this way, many circRNAs regulate ferroptosis by modulating ferroptosis-related genes by inhibiting the corresponding miRNAs. For example, circRNA IL4R (circIL4R) binds to and inhibits miR-541-3p to enhance GPX4, leading to the inhibition of ferroptosis in hepatocellular carcinoma [143]. CircLMO1 inhibits miR-4291 and elevates ACSL4 levels to induce ferroptosis in cervical cancer [140]. 

In addition, circRNAs were also found to regulate ferroptosis by directly binding to proteins. Circ-cIARS (hsa_circ_0008367) was found to bind to and block the RNA-binding protein ALKBH5, which is a negative regulator of ferritinophagy and ferroptosis [166]. CircEXOC5 can directly bind to RNA-binding protein polypyrimidine tract binding protein 1 (PTBP1) to enhance *ACSL4* mRNA stability, leading to ferroptosis in sepsis-induced acute lung injury [167]. It was reported that circRAPGEF5 interacts with and inhibits the splicing regulator RNA-binding protein fox-1 homolog 2 (RBFOX2) to confer ferroptosis resistance by modulating the alternative splicing of *TFRC* in endometrial cancer cells [168]. A recent study showed that circST6GALNAC6 interacts with small heat shock protein 1 (HSPB1) to inhibit its phosphorylation at the Ser-15 site, a phosphorylation site in the protective response to ferroptosis stress [169]. It was also found that circLRFN5 binds to PRRX2 protein and promotes its degradation, leading to the downregulation of GCH1 and inducing ferroptosis [170]. Moreover, circRNA circ101093 (cir93) was reported to interact with and increase fatty-acid-binding protein 3 (FABP3), which enhances the absorption and usage of AA to inhibit lipid peroxidation and ferroptosis [171]. 

Currently, circRNAs are reported to regulate ferroptosis in the above-mentioned manners, although they also show protein-coding and transcriptional regulation abilities. 

### 3.4. PiRNAs in Ferroptosis

PiRNAs are also a type of regulatory ncRNAs. They are small ncRNAs, which are different from miRNAs in that they are larger, lack sequence conservation, and are more complex [172]. They can combine with piwi proteins to make up a piRNA/piwi complex, which can cause gene silencing via interacting with a target transcript [173]. 

Few studies on piRNAs and ferroptosis have been reported. It was found in breast cancer cells that piR-36712 inhibits SEPW1 expression by binding to *SEPW1P* (a retroprocessed pseudogene of *SEPW1*) RNA, blocking its competition with *SEPW1* mRNA for miR-7 and miR-324, and it subsequently suppresses the ubiquitination of p53, enhancing the levels of p53 and its target p21 [174]. Since both p53 and p21 are regulators of ferroptosis [117], it is not surprising that piR-36712 may also be involved in the regulation of ferroptosis. Another study in prostate cancer showed that piR-31470 forms a complex with piwi-like RNA-mediated gene silencing 4 (PIWIL4) and then recruits DNMT1, DNA methyltransferase 3α, and methyl-CpG binding domain protein 2 to initiate and maintain the hypermethylation and inactivation of *glutathione S-transferase P1* (*GSTP1*) [175]. Related studies have indicated that *GSTP1* inactivation inhibits tumor cells from evading ferroptosis, leading to tumor growth [176], suggesting that piR-31470 may suppress ferroptosis through the inactivation of GSTP1.

### 3.5. Structural ncRNAs in Ferroptosis

In addition to regulatory ncRNAs, structural ncRNAs, such as tRNAs and rRNAs, are also involved in the regulation of ferroptosis. 

TRNAs are typically from 76 to 90 nucleotides in length (in eukaryotes) and contribute to protein synthesis and serve as the physical link between mRNAs and the amino acid sequence of proteins [177]. TRNAs are required in the synthesis of ferroptosis-associated proteins; thus, changes in tRNAs may alter the expression of these proteins and then influence ferroptosis. Interestingly, studies have also found that the mutation of tRNA results in a decrease in selenoprotein expression, except for GPX4 and GPX1, and weak ferroptosis alteration [178,179,180]. Moreover, it was found that a loss of cysteinyl-tRNA synthetase reduces GSH synthesis by inhibiting trans-sulfuration and decreasing cysteine levels, leading to the suppression of erastin-induced ferroptosis [79]. It seems that tRNAs decrease GSH synthesis and increase ferroptosis without modulating GPX4. On the contrary, it was also reported that Queuine-modified tRNAs promote antioxidant defenses by activating catalase, SOD, GPX, and GSH reductase [181], and the deletion of the selenocysteine-tRNA gene leads to the accumulation of ROS [182], suggesting that tRNAs may inhibit ferroptosis by enhancing the antioxidant defense system’s abilities. 

RRNAs are the primary component of ribosomes. They are also a kind of ribozyme that carries out protein synthesis in ribosomes [183]. Interestingly, a highly conserved 18S rRNA binding site was identified within the 5′-UTR of human *NRF2* mRNA, which is required for internal translation initiation, suggesting that the 18S rRNA regulates NRF2 expression [184]. This is supported by another finding that mouse hepatoma cells with a 70% decrease in the 16S/18S rRNA ratio caused by long-term ethidium bromide treatment showed increased NRF2 expression [185]. Considering the important roles of NRF2 in ferroptosis, 18S rRNA may regulate ferroptosis by regulating NRF2 expression. On the other hand, nuclear mitotic apparatus protein (NuMA), which is downregulated in response to ROS, can bind to 18S and 28S rRNAs and localize to rDNA-promoter regions to promote nascent pre-rRNA synthesis [186]. It was also reported that treatment with iron chelator deferoxamine inhibited rRNA synthesis in leukemia HL-60 cells [187]. These findings suggest that rRNA synthesis may be regulated during ferroptosis. 

SnRNAs mediate post-transcriptional splicing in gene expression, whereas snoRNAs mediate modifications to rRNAs, tRNAs, and snRNAs. Both of them are also structural ncRNAs. However, there are currently no reports on the relationship between ferroptosis and snRNAs or snoRNAs. 

Taken together, the current research mainly focuses on the regulation of ferroptosis by regulatory ncRNAs, especially miRNAs, lncRNAs, and circRNAs. Further studies are needed to explore the functions of piRNAs, tRNA, rRNAs, snoRNAs, and snRNAs in ferroptosis.

## 4. ncRNAs Regulating Ferroptosis

MiRNAs, lncRNAs, and circRNAs are the major ncRNAs that have been reported to regulate ferroptosis. They were found to regulate ferroptosis by directly modulating enzymes and other proteins involved in iron metabolism, lipid metabolism, and antioxidant defense, or by modulating other ferroptosis-related genes and proteins (e.g., transcription factors) indirectly.

### 4.1. NcRNAs Regulating Ferroptosis through Iron Metabolism

As one of the important mechanisms of ferroptosis, iron metabolism is regulated by many ncRNAs, which also affect ferroptosis (Figure 2). MiR-545 was reported to bind to the mRNA of *TF* and repress its expression to inhibit iron uptake and ferroptosis [188]. MiR-214 and miR-367-3p were found to target and repress TFRC to inhibit iron uptake and ferroptosis [138,189]. MiR-210-3p, which is enriched in hypoxia-conditioned cardiac microvascular endothelial cell-derived exosomes, also inhibits TFRC expression and attenuates erastin-induced myocardial cell ferroptosis [190]. LncPVT1, which competes with miR-214 in mouse brain I/R models, and lncRNA LINC00597, which competes with miR-367-3p in lung cancer cells, can increase TFRC expression to promote iron uptake and ferroptosis [138,189]. Moreover, circRAPGEF5 was reported to change the splicing of *TFRC* by binding to and inhibiting RBFOX2 and then inhibiting iron uptake to repress the ferroptosis of endometrial cancer cells [168]. 

Furthermore, miR-124-3p enriched in HO-1-modified bone marrow mesenchymal-stem cell-derived exosomes inhibits STEAP3 by directly interacting with its mRNA, suppressing the hypoxia/reoxygenation (H/R)-induced ferroptosis of IAR20 (normal rat hepatocyte cell line) and LO2 (human fetal hepatocyte cell line) by blocking the reduction of Fe^3+^ to Fe^2+^ and decreasing free Fe^2+^ levels [191]. MiR-375-3p targets DMT1 and downregulates its expression to inhibit ferroptosis by blocking the release of Fe^2+^ to the free iron pool [192]. MiR-23a-3p, which is carried by the exosome from human umbilical cord blood-derived mesenchymal stem cells, targets and represses DMT1 expression to inhibit the ferroptosis of myocardial cells isolated from mouse models for acute myocardial infarction (AMI) and cardiomyocyte hypoxia injury [193]. CircST6GALNAC6 interacts with HSPB1 to inhibit its phosphorylation at Serine 15, leading to the promotion of iron uptake and ferroptosis [169]. MiR-7-5p attenuates mitoferrin-1 to block mitochondrial iron accumulation, leading to ferroptosis suppression and radiation resistance in cancer cell lines [194]. 

MiR-124, miR-147a, miR-4735-3p, and miR-302a-3p target FPN1 and block iron export to facilitate ferroptosis in neuronal cells [141], lung cancer cells [195], clear cell renal cell carcinoma cells [196], and glioblastoma cells [197]; miR-761 reduces hepcidin levels to suppress FPN1 degradation in the liver, leading to a decrease in iron deposition and ferroptosis [198]; and miR-30b-5p represses Pax3 (a transcription factor) to downregulate *FPN1* transcription, thus inducing the ferroptosis of trophoblasts, leading to preeclampsia [142]. LncMAFG-AS1 binds to and stabilizes PCBP2 by the recruitment of deubiquitinase ubiquitin carboxyl-terminal hydrolase isozyme L5 (UCHL5) and then transports iron to FPN1 and suppresses ferroptosis [199]. MiR-129-5p targets and represses PROM2 to inhibit iron export, eventually promoting ferroptosis in non-small-cell lung cancer [200], whereas lncRP11-89 inhibits miR-129-5p to increase PROM2 expression, leading to the suppression of ferroptosis in bladder cancer [200].

MiR-335, miR-224-5p, and miR-19b-3p directly reduce FTH1 expression by binding to its mRNA, leading to an increase in free iron and the promotion of ferroptosis in Parkinson’s disease [201], heart failure [202], and lung cancer [203]. The inhibition of miR-224-5p by circSnx12 and the inhibition of miR-19b-3p by lncH19 elevate FTH1 levels to repress curcumenol-induced ferroptosis in heart failure [202] and lung cancer [203]. LncTUG1 targets MYC-associated zinc finger protein (MAZ) to reduce *FTH1* expression and then enhances DHA-induced ferroptosis in glioma cells [204]. In gestational diabetes mellitus (GDM), circHIPK3 blocks miR-1278 to enhance DNMT1 expression and facilitates ferroptosis by inducing NCOA4-mediated ferritinophagy [30,205]. LncA2M-AS1 directly interacts with PCBP3, an iron chaperone, to promote ferroptosis in pancreatic cancer [206]. It was also reported that circ-cIARS binds to and blocks RNA-binding protein ALKBH5 to promote ferritinophagy and ferroptosis [166].

Additionally, miR-19a negatively regulates IRP2 to inhibit ferroptosis by modulating iron metabolism in colorectal cancer [207]. N6-methyladenosine-modified circSAV1 forms a complex with YTHDF1 and *IRP2*, which in turn facilitates *IRP2* translation and accelerates cigarette-smoke-extract-induced ferroptosis in chronic obstructive pulmonary disease [208]. CircBCAR3 inhibits miR-27a-3p by the competitive RNA mechanism to upregulate transportin-1 (TNPO1) and then interacts with and inhibits carbonic anhydrase 9 to increase free iron accumulation by the upregulation of TFRC and the downregulation FTL and FTH1, eventually promoting ferroptosis [209,210].

### 4.2. NcRNAs Regulating Ferroptosis through Lipid Metabolism

Many ncRNAs were found to affect ferroptosis by regulating lipid metabolism (Figure 3). As an important marker and driver of ferroptosis, ACSL4 was reported to be targeted by a group of ncRNAs to regulate ferroptosis. MiR-34a-5p and miR-204-5p in prostate cancer [211,212], miR-424-5p in ovarian cancer [139], miR-670-3p in glioblastoma [213], miR-23a-3p in hepatocellular carcinoma [214], miR-4291 in cervical cancer [140], miR-1290 in non-small-cell lung cancer [215], miR-3098-3p in neuronal cells [216], miR-20a-5p in an acute kidney injury mice model and patients with delayed graft function [217], miR-29a-3p in hippocampal neurons after intracerebral hemorrhage [218], and miR-204 in HIV-1 Tat protein-exposed mouse primary microglial cells [219] directly target and limit ACSL4 expression to inhibit ferroptosis. It was also reported that miR-3173-5p carried by exosomes derived from cancer-associated fibroblasts represses ACSL4 to inhibit ferroptosis in GEM-resistant pancreatic cancer [220]. Moreover, lncNEAT1 competing with miR-34a-5p and miR-204-5p, circLMO1 competing with miR-4291, circSCN8A competing with miR-1290, and circCarm1 competing with miR-670-3p elevate ASCL4 levels and promote ferroptosis in prostate cancer [211,212], cervical cancer [140], non-small-cell lung cancer [215], and neuronal cells [216]. MiR-17-92 increases ACSL4 expression by directly targeting the zinc lipoprotein A20 to protect endothelial HUVEC cells from erastin-induced ferroptosis [221]. LncTUG1 carried by urine-derived stem-cell-derived exosomes reduces *ACSL4* levels by blocking serine/arginine splicing factor 1 (SRSF1) and inhibiting the H/R-induced ferroptosis of human proximal tubular epithelial cells in IRI-induced acute kidney injury [222]. CircEXOC5 binds to PTBP1 to promote *ACSL4* mRNA stability, leading to ferroptosis in sepsis-induced acute lung injury [167]. LncCBSLR decreased CBS levels to promote ACSL4 ubiquitination and degradation, thus protecting gastric cancer cells from ferroptosis [223].

In addition to targeting ACSL4, miR-7-5p targets and blocks ALOX12 to inhibit lipid peroxidation and ferroptosis, leading to radio-resistance of cancer cells [224]. MiR-522 in exosomes secreted by cancer-associated fibroblasts reduces ALOX15 expression by directly interacting with its mRNA, leading to the inhibition of lipid peroxidation and ferroptosis in gastric cancer [225]. MiR-18a directly binds to *ALOXE3* mRNA and represses its expression to inhibit ferroptosis in glioblastoma cells [226]. MiR-212-5p and miR-194-5p bind to the mRNA of *PTGS2* to repress its expression, thus inhibiting ferroptosis in neuronal death after traumatic brain injury [48] and protecting against temporal lobe epilepsy in young rats [227]. MiR-26a-5p carried by endothelial progenitor cell-derived exosomes also inhibits ferroptosis by targeting *PTGS2* mRNA to improve airway remodeling in chronic obstructive pulmonary disease [228]. 

Moreover, miR-423-5p targets and reduces SCD1 expression to promote the ferroptosis of colon cancer cells, which is inhibited by lncRNA LINC01606 [229]. CircZBTB46 also acts as an miRNA sponge to upregulate SCD1 and enhance RSL3-induced ferroptosis in acute myeloid leukemia (AML) cells [230]. In addition, exosomal lncFERO derived from gastric cancer cells was found to promote *SCD1* expression by directly interacting with *SCD1* mRNA and recruiting heterogeneous nuclear ribonucleoprotein A1 (hnRNPA1), which resulted in the suppression of ferroptosis in gastric cancer stem cells [231]. Exosomal cir93 from lung adenocarcinoma patients increases FABP3 to inhibit lipid peroxidation and ferroptosis [171]. In prostate cancer cells, miR-7 targets mTOR to suppress SREBP1, leading to the promotion of icariin- and curcumol-induced ferroptosis [232].

### 4.3. NcRNAs Regulating Ferroptosis through Antioxidant Defense

Approximately half of the current reported ferroptosis-regulating ncRNAs influence ferroptosis through antioxidant defense (Figure 4). SLC7A11 and GPX4, two important ferroptosis suppressors and markers, are targets not only for small molecules as ferroptosis inducers but also for a large group of ncRNAs regulating ferroptosis. MiR-409-3p, miR-515-5p, miR-375, miR-1261, miR-489-3p, miR-128-3p, miR-545-3p, miR-5096, miR-520d-5p, miR-125b-5p, miR-34c-3p, miR-143-3p, miR-26a-5p, miR-27b-3p, miR-587, miR-194-5p, miR-27a-3p, miR-520a-5p, miR-1184, miR-6077, miR-485-5p, miR-25-3p, miR-513a-3p, miR-206, and miR-431 have been reported to target and repress SLC7A11 to promote ferroptosis in cervical cancer [233], gastric cancer [148,234,235], liver cancer [236], prostate cancer [146], thyroid cancer [237], breast cancer [238], oral squamous cell carcinoma [239,240,241], kidney cancer [242,243], ovarian cancer [244,245], lung cancer [147,246,247,248,249], prostate cancer [250], esophageal squamous cell carcinoma [251], osteosarcoma [252], and colorectal cancer [253]. MiR-378a-3p, miR-27a, miR-144-3p, and exosomal miR-26b-5p from patients with acute myocardial infarction target and reduce SLC7A11 expression to induce ferroptosis in the IRI of the kidney [254], brain [255], and heart [256,257]. MiR-30b-5p directly binds to *SLC7A11* mRNA and represses its levels to induce the ferroptosis of trophoblasts under hypoxic conditions, leading to preeclampsia [142]. MiR-129-3p reduces the expression of SLC7A11 to induce ferroptosis under Se deficiency conditions, leading to liver damage [258]. MiR-16-5p inhibits SLC7A11 to promote ferroptosis in adriamycin-induced cardiomyocyte injury [259]. Exosomal miR-23a-3p derived from cardiac fibroblasts inhibits SLC7A11 expression to promote ferroptosis in atrial fibrillation [260]. It was also reported that miR-30b-5p reduces Pax3 levels to transcriptionally repress the expression of SLC7A11 to promote the hypoxia-induced ferroptosis of trophoblasts during preeclampsia [142]. MiR-367-3p carried by bone marrow mesenchymal stem cells (BMSCs)-derived exosomes targets an enhancer of zeste homolog 2 (EZH2) and restrains EZH2 expression, thus elevating SLC7A11 levels indirectly, and thus inhibiting the erastin-induced ferroptosis of microglia [261]. Functioning as competitive RNA, lncOIP5-AS1 inhibits miR-128-3p, lncSLC16A1-AS1 inhibits miR-143-3p, lncSNHG6 inhibits miR-26a-5p, lncCASC19/lncCYTOR/lncPVT1/lncRNA-LINC00997 inhibits miR-27b-3p, lncADAMT inhibits miRNA-587, lncBBOX1-AS1 inhibits miR-513a-3p, lncSNHG14 inhibits miR-206, circSnx12 inhibits miR-194-5p, circEPSTI1 inhibits miR-375/miR-409-3p/miR-515-5p, circRNA circ0097009 inhibits miR-1261, circRNA circ0067934 inhibits miR-545-3p, circFNDC3B inhibits miR-520d-5p, circFOXP1 inhibits miR-520a-5p, circP4H inhibits miR-1184, circRNA circ0070440 inhibits miR-485-5p, circRPPH1 inhibits miR-375, and circSTIL inhibits miR-431 to increase SCL7A11 levels, resulting in the inhibition of ferroptosis and the progression of prostate cancer [146], kidney cancer [242,243], ovarian cancer [244,245], esophageal squamous cell carcinoma [251], osteosarcoma [252], cervical cancer [233], liver cancer [236], thyroid cancer [237], oral squamous cell carcinoma [239], lung cancer [246,248,249], gastric cancer [234], and colorectal cancer [253]. Moreover, lncHEPFAL reduces SLC7A11 by promoting its ubiquitination and degradation, eventually enhancing erastin-induced ferroptosis in hepatocellular carcinoma [155]. On the other hand, lncRNA LINC00578 also inhibits SLC7A11 ubiquitination by binding to UBE2K to inhibit ferroptosis in pancreatic cancer [262]. CircBGN directly binds to OTUB1 and SLC7A11, enhancing OTUB1-mediated SLC7A11 deubiquitination to inhibit ferroptosis in breast cancer [263]. Furthermore, lncAGAP2-AS1 stabilizes *SLC7A11* mRNA via the IGF2BP2 pathway to suppress ferroptosis in melanoma cells [264]. LncPMAN stabilizes SLC7A11 mRNA by recruiting ELAVL1 to the cytoplasm to inhibit the ferroptosis of gastric cancer cells [154]. LncSLC7A11-AS1 inhibits SLC7A11 expression to induce ferroptosis in ovarian cancer [265]. Additionally, lncRNA LINC00618 attenuates the expression of lymphoid-specific helicase (LSH), which can bind to the promoter regions of SLC7A11 to promote its transcription. Therefore, lncRNA LINC00618 represses SLC7A11 expression to accelerate ferroptosis in human leukemia [266]. 

Similarly, GPX4 was reported to be targeted to induce ferroptosis by miR-541-3p and miR-214-3p in liver cancer [143,267], miR-324-3p in breast cancer [144] and kidney cancer [145], miR-1231 in thyroid cancer [268], miR-1287-5p/miR-744-5p/miR-615-3p in lung cancer [269,270], miR-1287-5p in osteosarcoma [271], miR-15a in prostate cancer [272], miR-193a-5p in cervical cancer [273], miR-3202 in cholangiocarcinoma [274], as well as by miR-182-5p, miR-1224, miR-15a-5p, miR-135b-3p in the IRI of kidney injury [254] and heart [275,276,277], miR-23a-3p in intracerebral hemorrhage [278], miR-188 in diabetic nephropathy [279], miR-761 in liver dysfunction of patients with polycystic ovary syndrome [198], exosomal miR-208a/b secreted from hypoxia-induced cardiomyocytes in cardiomyocytes and cardiac fibroblasts [280], and exosomal miR-700-5p from hypoxia-pretreated adipose-derived stem cells in UV-light-induced skin injury [281]. The inhibition of miR-214-3p by lncPVT1 [267] and the inhibition of miR-541-3p by circIL4R [143] in liver cancer, the inhibition of miR-3202 by lncRNA LINC00976 in cholangiocarcinoma [274], the inhibition of miR-1231 by circKIF4A in thyroid cancer [268], the inhibition of miR-1287-5p by circDTL in lung cancer [270], the inhibition of miR-193a-5p by circACAP2 in cervical cancer [273], and the inhibition of miR-188 by circRNA circ0000309 in diabetic nephropathy [279] elevate GPX4 levels to suppress ferroptosis. Moreover, circIDE inhibits miR-19b-3p as a sponge to elevate the expression of RBMS1, which in turn reduces the stability of *GPX4* mRNA to facilitate ferroptosis in hepatocellular carcinoma [282]. LncRNA LINC01134 recruits NRF2 to the promoter region of *GPX4* to enhance *GPX4* transcription, leading to the suppression of ferroptosis in hepatocellular carcinoma [283]. MiR-4715-3p targets and represses Aurora kinase A (AURKA) to reduce GPX4 protein levels in an unknown mechanism and then induces ferroptosis in upper gastrointestinal adenocarcinoma [284]. 

In addition to SLC7A11 and GPX4, ncRNAs regulate other members of the GPX4-GSH system to affect ferroptosis. MiR-6852 targets and represses CBS to reduce cysteine and GSH synthesis, leading to the promotion of ferroptosis, whereas lncRNA LINC00336 acts as a sponge to block miR-6852 and inhibit ferroptosis in lung cancer [285]. MiR-145-5p directly binds to the mRNA of *GCLM* (a subunit of GCL) and represses its expression, resulting in a decrease in GSH levels and the induction of ferroptosis in prolactinomas, whereas circOMA1 inhibits miR-145-5p in a competitive manner to suppress ferroptosis [286]. It was also found that exosomal miR-760-3p from adipose-derived mesenchymal stem cells targets and represses CHAC1 to reduce GSH and attenuate ferroptosis in neurons [287]. 

NcRNAs also modulate ferroptosis by regulating glutaminolysis in two ways. MiR-338-3p targeting SLC1A5 in retinal pigment epithelium cells [87] and miR-299-3p targeting SLC38A1 in lung cancer cells [85] reduce glutamine absorption and GSH levels to promote ferroptosis, whereas lncOGFRP1 attenuates ferroptosis by blocking miR-299-3p in lung cancer cells [85]. MiR-9 and miR-2115-3p bind to the mRNA of *GOT1* to repress its expression, leading to suppression of ferroptosis probably by the elevation of glutamate and GSH levels in melanoma cells and a preeclampsia model, respectively [288,289]. On the contrary, miR-137 targeting SLC1A5 in melanoma cells [89], miR-150-5p targeting SLC38A1 during pulmonary fibrosis [92], and miR-15b-5p and miR-190a-5p targeting GLS2 in pancreatic β-cells and rat cardiomyocyte cells [93,95] inhibit ferroptosis by blocking glutaminolysis and lipid synthesis, whereas lncZFAS1 acts as a competitive endogenous RNA to block miR-150-5p to facilitate ferroptosis during pulmonary fibrosis [92]. Moreover, lncATXN8OS stabilizes GLS2 mRNA to facilitate ferroptosis in glioma [94]. LncSnhg7 interacts with T-box transcription factor 5 (Tbx5) to transcriptionally induce GLS2 expression, resulting in the promotion of cardiomyocyte ferroptosis in cardiac hypertrophy [96]. 

Additionally, ncRNAs regulate ferroptosis via antioxidant systems other than the GPX4-GSH system. MiR-1228, miR-5627-5p, and miR-672-3p target and suppress FSP1 to promote ferroptosis in breast cancer cells [290], in neuronal cells [291], and in rats with contusive spinal cord injury [292], whereas circGFRA and lncGm36569 carried by mesenchymal-stem-cell-derived exosomes promote ferroptosis by directly targeting miR-1228 and miR-5627-5p, respectively [290,291]. It was also reported that exosomal miR-4443 from cisplatin-resistant lung cancer tissue inhibits ferroptosis by directly targeting METLL3, which regulates the m6A modification of *FSP1* mRNA [293]. Moreover, LncTMEM161B-AS1 directly blocks miR-27a-3p to increase the expression of its target GCH1, leading to the suppression of ferroptosis in esophageal cancer [293]. CircLRFN5 interacts with PRRX2 and promotes its ubiquitination and proteasomal degradation, transcriptionally reducing PRRX2-mediated GCH1 expression, resulting in the induction of ferroptosis in glioma [170]. LncGABPB1-AS1 downregulates PRDX5 by blocking GABPB1 translation to promote the erastin-induced ferroptosis of hepatocellular carcinoma cells [106]. LncNEAT1 competitively binds to miR-362-3p to increase the expression of myo-inositol oxygenase (MIOX), a non-heme-iron enzyme, to promote ROS production and the ferroptosis of hepatocellular carcinoma cells [294]. MiR-214-3p targets malic enzyme 2 (ME2) to suppress the cellular antioxidant capacity and promote ferroptosis in neonatal rat cardiomyocytes [295]. Cancer-associated fibroblast-derived exosomal lncDACT3-AS1 inhibits miR-181a-5p to elevate the levels of its target sirtuin 1 (SIRT1), leading to an increase in antioxidant capacity to inhibit ferroptosis in gastric cancer [296]. 

### 4.4. NcRNAs Regulating Ferroptosis through Other Ferroptosis Regulators

There are also some ncRNAs that regulate ferroptosis through other ferroptosis regulators, e.g., transcription factors. MiR-365a-3p directly targets and suppresses NRF2 to promote ferroptosis in non-small-cell lung cancer [297]. LncMT1DP facilitates erastin-induced ferroptosis by stabilizing miR-365a-3p [297]. MiR-6077 and exosomal miR-125b-5p from adipose-derived stem cells directly repress KEAP1, a negative regulator of NRF that regulates its ubiquitination and degradation, to increase the expression and nucleus translocation of NRF2, leading to the alleviation of ferroptosis in lung adenocarcinoma cells [247] and pulmonary microvascular endothelial cells [298]. LncGMDS-AS1 and lncRNA LINC01128 promote ferroptosis by competitively blocking miR-6077 [247]. Moreover, lncRNA LINC00239 interacts with the NRF2 binding site of KEAP1 to increase NRF2 levels by ubiquitination inhibition and then inhibits ferroptosis in colorectal cancer [299]. Additionally, miR-130b-3p targets and represses Dickkopf1 to activate NRF2, leading to ferroptosis suppression in melanoma [300].

In addition to NRF2, miR-214 and let-7b-5p target and repress p53 to inhibit ferroptosis in brain I/R and acute myeloid leukemia, which are directly blocked by lncPVT1 and circKDM4C, respectively [138,301]. LncMeg3 directly interacts with p53 and enhances its stability and transcriptional activity, mediating ferroptosis induced by oxygen and glucose deprivation combined with hyperglycemia in rat brain microvascular endothelial cells [302,303,304]. LncP53RRA can interact with Ras GTPase-activating protein-binding protein 1 (G3BP1) to sequester p53 in the nucleus, thus promoting ferroptosis [156]. LncPELATON forms a complex with p53 and the RNA-binding protein EIF4A3 to inhibit ferroptosis in glioblastoma [305]. 

Moreover, miR-221-3p was reported to target ATF3 to transcriptionally induce the expression of *GPX4* and *HRD1* and then promote the ubiquitination and degradation of ACSL4 by HRD1, resulting in ferroptosis suppression in gastric cancer cells [306]. LncDLEU1 binds to ZFP36 and facilitates the degradation of *ATF3* mRNA by ZFP36, thus upregulating the expression of SLC7A11 to attenuate erastin-induced ferroptosis in glioblastoma [307]. MiR-214 and miR-3200-5p directly repress ATF4 expression to enhance ferroptosis in hepatocellular carcinoma cells [308,309], whereas LncHULC increases ATF4 levels by competing with miR-3200-5p and inhibits ferroptosis [308]. CircRHOT1 directly inhibits miR-106a-5p to increase its target STAT3, resulting in ferroptosis inhibition in breast cancer cells [310]. It was also reported that miR-125b-5p targets and represses STAT3 to induce ferroptosis in gastric cancer [311]. CircRNA circ0008035 increases the expression of E2F transcription factor 7 (E2F7) by blocking miR-302a, inhibiting dexmedetomidine-induced ferroptosis in gastric cancer cells [312]. CircASAP2 blocks miR-770-5p to enhance the expression of SRY-Box transcription factor 2 (SOX2) and then induces *SLC7A11* expression to suppress ferroptosis in diabetic nephropathy [313]. Moreover, miR-7-5p upregulates *HIF-1α* expression by an unknown mechanism to inhibit ferroptosis in radioresistant HeLa (cervical carcinoma) and SAS (human tongue carcinoma) cell lines [224].

NcRNAs also regulate ferroptosis by other signaling pathways. CircTTBK2 inhibits miR-761 to modulate its target ITGB8, a subunit of integrin beta, to reduce ferroptosis in glioma [314]. CircRNA circ0007142 was found to block the effect of miR-874-3p on its target GDPD5, leading to ferroptosis suppression in colorectal cancer [315]. LncMEG8 inhibits miR-497-5p to increase NOTCH2, leading to the inhibition of ferroptosis by elevating SLC7A11 and GPX4 levels [316]. CircRNA circ0000745 competitively inhibits miR-494-3p to upregulate neuroepithelial cell transforming 1 (NET1), a guanine nucleotide exchange factor of RhoA, to suppress ferroptosis in acute lymphoblastic leukemia cells [317]. CircPVT1 competes with miR-30a-5p to elevate the levels of its target Frizzled3, activating Wnt/β-Catenin signaling to inhibit ferroptosis [318]. LncA2M-AS1 forms a complex with PCBP3 to facilitate p38 activation and inhibit the AKT-mTOR signaling pathway, contributing to the promotion of ferroptosis [206]. In addition, circABCB10 acts as a sponge for miR-326 to elevate CCL5 levels and attenuate the ferroptosis of rectal cancer cells [319]. LncRNA LINC00460 blocks miR-320a to enhance the expression of myelin and lymphocyte protein 2 (MAL2), inhibiting ferroptosis in breast cancer [320].

## 5. Conclusions and Perspectives

Ferroptosis is a novel type of regulated cell death that was first proposed in 2012 [3]. Ferroptosis has been found in diverse species, including humans, other mammals and vertebrates, invertebrates, plants, yeast, and bacteria [321,322,323,324]. Substantial studies have focused on exploring the mechanisms of ferroptosis and understanding how it is regulated in different cells, especially in humans and other mammals. Collectively, these studies have demonstrated that ferroptosis results from lipid peroxidation caused by the dysregulation of iron metabolism, lipid metabolism, and antioxidant defense, which is controlled by regulating enzymes and other proteins involved in these processes at the transcriptional, post-transcriptional, translational, and post-translational levels. NcRNAs have been proven to regulate gene expression in various manners. In recent years, numerous ncRNAs, including miRNAs, lncRNAs, and circRNAs, have been reported to regulate ferroptosis, which have been summarized in this review. These ncRNAs can directly target ferroptosis-related enzymes or proteins involved in iron metabolism, lipid metabolism, and antioxidant defense, or they can indirectly target other regulators of ferroptosis, such as transcription factors. Some ncRNAs also act as molecular sponges or form complexes with other ncRNAs to exert their functions, which adds another level of regulation on ferroptosis. Moreover, because of the differential expression of ncRNAs in different cells, these ncRNAs may affect the regulation of ferroptosis in a cell-type-dependent or tissue-type-dependent manner. Therefore, although great progress has been made in studying the molecular mechanism of ncRNA regulation of ferroptosis, a deeper understanding of the mechanisms of how ncRNAs regulate ferroptosis in cell- or tissue-specific manners is still required. Moreover, ferroptosis is also reported in fish, invertebrates, plants, yeast, and bacteria [321,322,323,324], which often contain different ncRNAs in their genomes. Few studies have reported the regulation of ferroptosis by ncRNAs in these species, which may be different from our current knowledge in humans and other mammals.

Ferroptosis functions are intricately involved in numerous physiological processes and various diseases, such as cancer, neurodegeneration disease, cardiovascular diseases, and the IRI of different organs such as the heart, brain, kidney, and liver [9,10,11,12,13,14]. Therefore, some ncRNAs regulating ferroptosis, including miRNAs, lncRNAs, and circRNAs, have the potential to be novel therapeutic methods and diagnostic biomarkers for these diseases, especially for cancer. On the one hand, more and more studies suggest that ncRNAs can be used as a biomarker to predict tumor progression and clinical prognosis. However, the most important ncRNAs associated with specific types of tumors are still not identified, which limits the application of ncRNAs as diagnostic biomarkers. On the other hand, according to studies on ncRNA-regulated ferroptosis, the ferroptosis process can be altered via targeting ncRNAs, thus affecting the ferroptosis-related pathological process, e.g., cell proliferation and chemoresistance for cancer cells. Current inducers and inhibitors of ferroptosis mainly target the proteins involved in the core mechanism of ferroptosis, which may induce metabolic dysregulation in normal cells and cause unexpected side effects in addition to regulating ferroptosis. Due to their cell- or tissue-specific manners of regulating ferroptosis, ncRNAs may be better targets for therapeutic methods. Furthermore, because of the heterogeneity of gene expression on a per individual basis, ncRNA-associated therapy and biomarkers can be applied to support the personalized treatment of ferroptosis-related disease. Additionally, ncRNAs are also promising therapeutic agents for ferroptosis-related diseases. Indeed, ncRNAs have already shown their effect of regulating ferroptosis in vitro. Their low in vivo bioavailability has limited their clinical application. In general, three strategies have been proposed for ncRNA-based therapies, including nanoparticles, ncRNA modification, and oncolytic adenovirus strategy [325]. Although no project has entered the clinical trial stage, progress has been achieved in the clinical application of ncRNA to regulate ferroptosis. Moreover, since the regulation of ferroptosis by ncRNAs derived from exosomes has been reported in many studies, exosomes can be used as excellent ncRNA transporters to induce or inhibit ferroptosis in cells. With more extensive research, loading selected ncRNAs into exosomes to induce ferroptosis may create novel opportunities for ncRNA-based therapies.

Taken together, ncRNAs form a complex network to regulate ferroptosis via proteins or genes involved in the core mechanism of ferroptosis or its regulators. This review has summarized the regulatory roles of several types of ncRNAs in ferroptosis, which are beneficial for understanding the pathogenesis of ferroptosis-related diseases. And these ferroptosis-related ncRNAs have great potential to act as therapeutic targets or agents for novel therapeutic methods, as well as diagnostic biomarkers, for these diseases, especially for cancers.

## Figures and Tables

**Figure 1 ijms-24-13336-f001:**
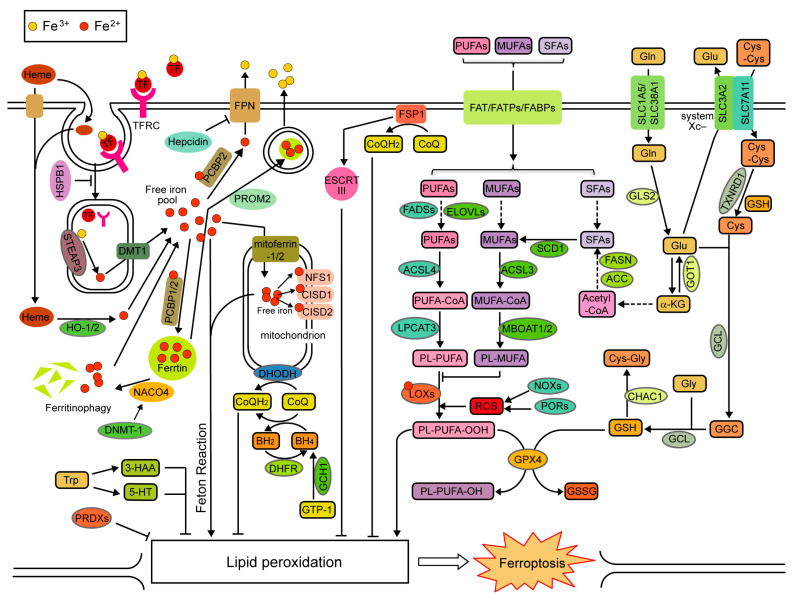
Schematic illustration of the core mechanisms of ferroptosis. Ferroptosis is regulated by iron metabolism, lipid metabolism, and antioxidant defense systems. α-KG: α-ketoglutarate; BH_2_: dihydrobiopterin; Cys: cysteine; Cys-Cys: cystine; Cys-Gly: cysteinylglycine; Gln: glutamine; GGC: γ-glutamyl cysteine; GSSG: glutathione oxidized; Glu: glutamate; Gly: glycine; and Trp: tryptophan.

**Figure 2 ijms-24-13336-f002:**
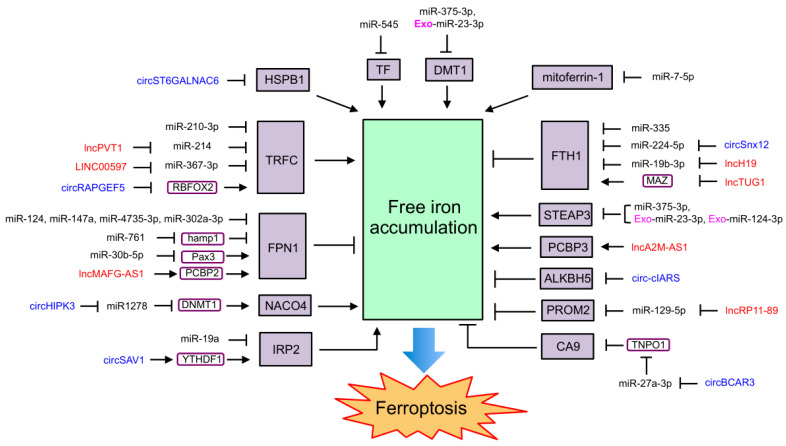
ncRNAs regulating ferroptosis through iron metabolism. MiRNAs are unboxed text in black. LncRNAs are unboxed text in red. CircRNAs are unboxed text in blue. Exosomal ncRNAs are marked with an “Exo-” in pink. Genes and proteins that directly involved in ferroptosis are black text boxed in black and shaded in purple. Other genes and proteins mediated the function of ncRNAs on ferroptosis are black text boxed in deep purple.

**Figure 3 ijms-24-13336-f003:**
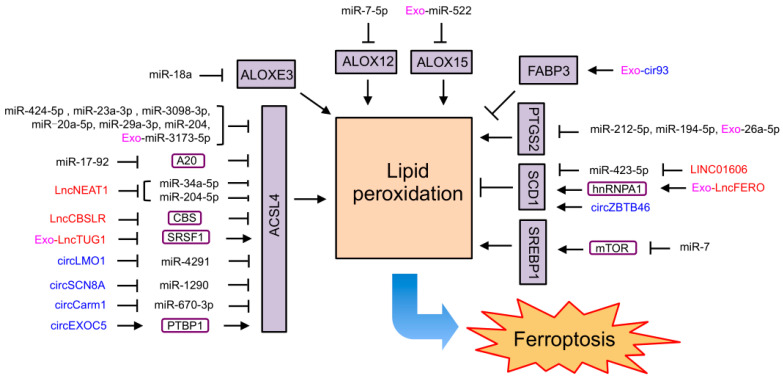
ncRNAs regulating ferroptosis through lipid metabolism. MiRNAs are unboxed text in black. LncRNAs are unboxed text in red. CircRNAs are unboxed text in blue. Exosomal ncRNAs are marked with an “Exo-” in pink. Genes and proteins that directly involved in ferroptosis are black text boxed in black and shaded in purple. Other genes and proteins mediated the function of ncRNAs on ferroptosis are black text boxed in deep purple.

**Figure 4 ijms-24-13336-f004:**
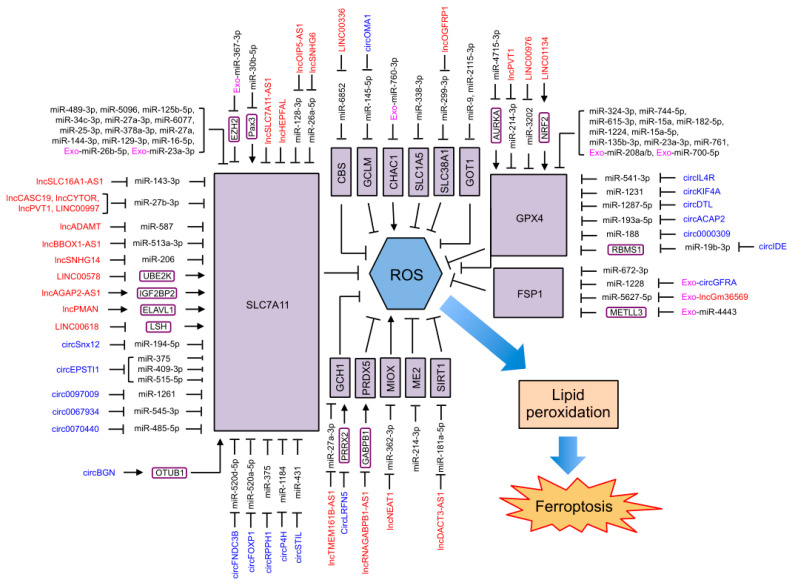
ncRNAs regulating ferroptosis through antioxidant defense. MiRNAs are unboxed text in black. LncRNAs are unboxed text in red. CircRNAs are unboxed text in blue. Exosomal ncRNAs are marked with an “Exo-” in pink. Genes and proteins that directly involved in ferroptosis are black text boxed in black and shaded in purple. Other genes and proteins mediated the function of ncRNAs on ferroptosis are black text boxed in deep purple.

## Data Availability

Not applicable.

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
