# Peer review of "The Regulation of Ferroptosis by Noncoding RNAs"

_ijms, 2023, doi:10.3390/ijms241713336_

Round 1
Reviewer 1 Report
Int. J. Mol. Sci.- 2561692_2023The manuscript entitled “The regulation of ferroptosis by non-coding RNAs” by Xiangnan Zheng and Cen Zhang submitted to section ‘Molecular Endocrinology and Metabolism’ for the Special Issue Recent Advance on Iron Metabolism, Ferritin and Hepcidin Research 2.0.
The manuscript provides a review and is devoted to a very actively developing field of oncology, namely, programmed cell death. In particular, the authors summarized and discuss the regulatory roles of several types of ncRNAs in ferroptosis which are beneficial for understanding the pathogenesis of ferroptosis-related diseases.
In recent years, ferroptosis has become a research hotspot in programmed cell death.
The current attention in this field mainly focuses on potential regulatory mechanism and pathways including key ferroptosis-related genes/molecules.
Overall, although much progress has been made, the research on ferroptosis is still at an early stage.
The authors carried out a very serious and thorough analysis, which seems important and useful for the development of this area and further progress in the creation of new drugs and diagnostic biomarkers.
The literature material is well organized. This review cites 320 sources, predominantly research articles and reviews.
These materials could be useful to specialists in the field of medicinal chemistry and pharmacology. They will also be of interest to biochemists and chemists working with bioactive compounds.
In general, the manuscript is well prepared and carefully verified.
The manuscript does not raise any objections, and can be published in Int. J. Mol. Sci. after minor revision.
I would like to note the excellently prepared ‘ncRNAs regulating ferroptosis’ and ‘Conclusions and perspectives’ sections.
Several details and inaccuracies should be noted.
1. References. Authors should carefully check the References, and especially the generally accepted journal abbreviation.
English is good.
Author Response
Dear Reviewer,
Thank you for your time, and positive and constructive comments on our manuscript. We have read your comments and addressed the critiques for this revision.
According to your suggestion, we checked the references carefully, changed all the journal abbreviations to the same form as PubMed, and corrected the errors we found. We also asked a native English speaking colleague to check our manuscript and corrected some errors according to his suggestion. We hope that with these changes, our manuscript is acceptable for publication.
Again, we want to thank you for the very nice comments, very constructive suggestions, and great efforts to improve our manuscript.
Thank you very much!
Sincerely yours,
Cen Zhang, Ph.D.
Professor,
College of Biological Science and Engineering,
Fuzhou University,
No. 2 Xueyuan Road
Fuzhou, Fujian 350108, China
Reviewer 2 Report
Review article on "The regulation of ferroptosis by non-coding RNAs" is overall a good read. The manuscript effectively addresses a wide array of topics related to the intricate relationship between non-coding RNAs and ferroptosis. The depth of coverage is commendable, and it is evident that the authors have undertaken a thorough exploration of the subject matter.
The article offers a wealth of valuable insights into the regulatory roles of non-coding RNAs in the context of ferroptosis. The inclusion of various relevant points and aspects strengthens the credibility of the review. The authors' diligence in compiling information from diverse sources contributes to the overall richness of the content.
However, it is important to note that some sections of the article present challenges in terms of readability. The complexity of certain paragraphs could hinder a smooth comprehension of the material. To enhance the accessibility of the article to a broader readership, I suggest that the authors consider revising these intricate passages, perhaps by breaking down complex concepts into more manageable segments or providing clearer explanations.
Author Response
Dear Reviewer,
Thank you very much for your careful review, and positive, insightful, and constructive comments on our manuscript. We have revised our manuscript accordingly and appropriate changes have been incorporated into our manuscript.
We added some subtitles and rearranged some paragraphs for section 3 to make it more clearly for different ncRNA types. We also rearranged some paragraphs and sentences for section 4 to make the logic similar with the section 2. We think these may help to improve the readability of our manuscript. We hope that you find our responses acceptable and approve this revised manuscript for publication
Again, we want to thank you for the very nice comments, very constructive suggestions, and great efforts to improve our manuscript.
Thank you very much!
Sincerely yours,
Cen Zhang, Ph.D.
Professor,
College of Biological Science and Engineering,
Fuzhou University,
No. 2 Xueyuan Road
Fuzhou, Fujian 350108, China
Reviewer 3 Report
ijms-2561692 The regulation of ferroptosis by non-coding RNAsAuthors described in mentioned manuscript the regulation of ferroptosis by complex network of protein and genes formed by ncRNAs. The review summarized the regulatory roles of several types of ncRNAs in ferroptosis which are beneficial for understanding the pathogenesis of ferroptosis-related diseases. Article is well-written, and structured. Authors starts from briefly description of ferroptosis process, then they put some information about biochemical characterisation of ferroptosis (abnormal iron accumulation, increased lipid peroxidation, dysregulation of antioxidant defense), and next classification and function of ncRNAs in ferroptosis.
I have one suggestion. The ferroptosis is newly described form of cells death, mostly in human and mammals, are there any information about evolutionary conservatism of this proces, I mean if there are some information about presence and function of this process in for examples in Invertebrates.
Author Response
Dear Reviewer,
Thank you very much for your careful review, and positive, insightful, and constructive comments on our manuscript. We have revised our manuscript accordingly and appropriate changes have been incorporated into our manuscript.
Regarding to your suggestion, ferroptosis has been also found in fish, invertebrates, plants, yeast, and bacteria, which often contain different ncRNAs in their genomes. Currently no new mechanisms for ferroptosis involved in ncRNAs has been reported in these species. Major studies on ferroptosis in these species reported similar and conserved mechanism. It will be an interesting direction to find unconserved mechanism with different ncRNAs in these species. We have mentioned all these in the first paragraph of “Conclusions and perspectives” section. We hope that you find our responses acceptable and approve this revised manuscript for publication.
Again, we want to thank you for the very nice comments and great efforts to improve our manuscript.
Thank you very much!
Sincerely yours,
Cen Zhang, Ph.D.
Professor,
College of Biological Science and Engineering,
Fuzhou University,
No. 2 Xueyuan Road
Fuzhou, Fujian 350108, China